# Development and Validity of a General Nutrition Knowledge Questionnaire (GNKQ) for Chinese Adults

**DOI:** 10.3390/nu13124353

**Published:** 2021-12-03

**Authors:** Zhibing Gao, Fei Wu, Gaoyaxin Lv, Xiangling Zhuang, Guojie Ma

**Affiliations:** Shaanxi Key Laboratory of Behavior and Cognitive Neuroscience, School of Psychology, Shaanxi Normal University, Xi’an 710062, China; zb_gao@snnu.edu.cn (Z.G.); wuf@snnu.edu.cn (F.W.); lvgyx@snnu.edu.cn (G.L.); zhuangxl@snnu.edu.cn (X.Z.)

**Keywords:** general nutrition knowledge questionnaire, adults, reliability, validity

## Abstract

Nutrition knowledge refers to understanding concepts and processes related to nutrition and health, proven to be an essential determinant of healthy eating. However, partially due to the lack of nutrition knowledge and unhealthy eating patterns, more and more Chinese people face overweight, obesity, and a high risk of suffering from various chronic diseases. This study aimed to develop a general nutritional knowledge questionnaire (GNKQ) in a Chinese context to diagnose and improve nutrition knowledge education for Chinese people. The newly adapted questionnaire was based on the Turkey version of GNKQ, and absorbed dietary recommendations in a Chinese context. It was first validated by four nutrition experts, then tested by eleven volunteers (one public nutritionist, one preventive medicine graduate student, and nine psychology graduate students). Finally, the questionnaire was tested by 278 participants, including 175 adults, to determine internal consistency, content validity, and convergent validity. Moreover, the construct validity was evaluated by comparing the differences between 50 students in nutrition-related majors and 53 students in nutrition-unrelated majors. The final Chinese version of GNKQ kept 32 questions with 68 items after deleting some questions based on item difficulty and discrimination. The data showed that the overall internal consistency coefficient was 0.885, and the test-retest reliability was 0.769, *p* < 0.001. Students majoring in nutrition had larger scores than in nutrition-unrelated majors. The convergent validity for each demographic variable was consistent with previous studies, such as larger nutrition knowledge scores for females and those with a higher education. Therefore, the revised Chinese version of GNKQ showed good reliability and validity, indicating that it could be an effective tool to assess the nutrition knowledge of Chinese adults.

## 1. Introduction

Nutrition knowledge refers to an understanding of concepts and processes related to nutrition and health [1]. It has been proved to be an important determinant of nutrition label use and food-related decision making [2,3]. Usually, people with more nutritional knowledge understand nutritional information better, and, therefore, usually have relatively healthy eating patterns, such as eating more fruits [4], vegetables [5], healthy foods [6], and less fat [7]. Though there are still lots of unanswered questions about the relationship between nutrition and health, compared with bad eating habits, such as consuming more high-fat, high-energy-density foods [8], those healthy dietary behaviors for the general population play a crucial role in preventing food-related chronic diseases such as obesity, ischemic heart disease, and cancer [9]. Therefore, diagnosing and improving people’s nutrition knowledge levels is critical for maintaining public health.

In China, food-related chronic diseases, such as obesity and vascular diseases have been rising in recent years [10,11]. These problems reveal that people’s nutrition knowledge is limited, and they cannot effectively use the *Dietary Guidelines for Chinese Residents* to select foods and restrict what they eat. In order to solve this problem, the Chinese government introduced ‘The Healthy China 2030 Plan’, with the goal of improving the health service and health literacy of Chinese people by 2030. To achieve this goal, nutritional knowledge should be repeatedly estimated using appropriate tools. However, a nutrition knowledge questionnaire has not been systematically studied or created for Chinese people. 

Parmenter and Wardle [12] first developed the general nutrition knowledge questionnaire (GNKQ) in the UK and proved its reliability and validity for UK people. The GNKQ consists of questions that examine (1) the understanding of current dietary recommendations; (2) the knowledge of food sources related to nutrients; (3) the use of dietary information to make food choices; and (4) the relationship between diet and diseases. The GNKQ was then adapted in Australia [13], Turkey [14], and Japan [15], according to different dietary guidelines and recommendations in their countries. Previous studies have revealed that the revised versions of GNKQ can effectively distinguish different nutrition knowledge levels [13] and predict food choice behavior [16]. It has also been observed that nutrition knowledge levels showed significant differences across different demographic variables, such as age, gender, and education level [17]. 

Even though the GNKQ [12] has been widely adapted and used in different countries, it has not been validated in a Chinese context. Since food patterns and nutrition recommendations vary between cultures, the adjustment of GNKQ for a Chinese context is also necessary. Previous studies modified GNKQ mainly through adding items about national dietary policies, replacing unusual foods with native foods, and testing its reliability and validity in a national sample [15]. Previous studies have shown that, compared with the dietary patterns in Western countries, the intake of some healthy foods and nutrients, such as fruits, milk, calcium, and omega-3 fatty acids is more inadequate in East Asia, such as China. However, the intake of some relatively unhealthy foods and nutrients, such as deeply processed meats, sugary beverages, and trans fatty acids is too much. It is alarming that China’s sodium intake (9.3 g daily per individual in 2020) is much higher than the recommended healthy intake standards [18,19,20,21]. Chinese traditional eating patterns and national dietary recommendations are quite different from Western countries, which should be emphasized in modifying the GNKQ to the Chinese context. However, there has been no study systematically elaborating on the Chinese version of GNKQ. Given that more than half of Chinese adults belong to the overweight and obesity group [18], developing a Chinese version of GNKQ seems necessary and urgent. This study aimed to develop a general nutrition knowledge questionnaire for Chinese adults (C-GNKQ) and test its reliability and validity. In addition, by testing its construct validity among college students, this study also aimed to distinguish the level of nutrition knowledge among young people with different nutrition educations.

## 2. Methods

### 2.1. Questionnaire Selection and Translation

The general nutrition knowledge questionnaire (GNKQ) has been most widely used to examine nutritional knowledge [12]. The early version of GNKQ has been successfully revised in Australia [13], Turkey [14], the United States [22], the United Kingdom [23], Uganda [24], Japan [15], Portugal [17], Italy [25], and South Africa [26]. Among them, in the Turkish version, the items of “recommended intake of salt”, “food containing trans fatty acids”, “iron utilization in plants compared with iron in animals”, and “the relationship between diet and disease’’ were added. In addition, closed questions were replaced with semi-open questions, such as “whether the following foods are eaten more or less.” The GNKQ was mainly revised based on the Turkish version because it was much more consistent with Chinese dietary guidelines. In Turkey, cardiovascular diseases, cancer, and diabetes cause 65% of deaths and nutrient deficiency and are major public health problems, thus the Turkish population is encouraged to consume foods rich in iron, iodine, and vitamin D [14]. However, according to China’s latest Scientific Research Report on Dietary Guidelines for Chinese Residents (2021), Chinese people do not currently suffer from significant deficiencies in the intake of the above nutrients. Even so, it should be noted that the people in both Turkey and China consume too many high-sodium foods, which in turn potentially leads to a higher risk of obesity, cardiovascular disease, and diabetes in both countries.

The original questionnaire of the Turkish version of GNKQ [14] was translated through the forward-and-back translation method independently by an undergraduate and a postgraduate who majored in dietary psychology. After a group discussion, the initial Chinese version of GNKQ was formed. Then, the questionnaire was translated back by a public nutritionist and a graduate student with an English translation major. Finally, all translators, back-translators, and an associate professor of dietary psychology confirmed the first draft of C-GNKQ after checking and discussing all translation differences.

#### 2.1.1. Generation of Items

According to the *Chinese Dietary Guidelines (2016)*, a public nutritionist replaced all Turkish foods with Chinese foods, based on the *Chinese Food Composition 6th Edition*. The first draft included 43 questions with 133 items. Then, we added 11 questions (28 items) to the test version of the GNKQ from the Knowledge Attitude Practice (KAP) [27]. These 11 questions were categorized as follows: *dietary recommendations* (4 questions, 13 items); *sources of nutrients* (6 questions, 14 items); *the diet-disease relationship* (1 question, 1 item). The content of these 11 questions mainly included what is mentioned in the Chinese Dietary Guidelines, such as types of essential nutrients, the recommended ratio of calorie intake for each meal, and which is more recommended for dark or light colored vegetables. After that, three public nutritionists and a nutrition professor checked whether these questions were suitable for measuring Chinese adults’ nutrition knowledge. Finally, 50 questions with 158 items were confirmed as the test version of C-GNKQ. Three answer types were used: yes, no, or do not know. All questions were closed and contained ‘unknown’ or ‘uncertain’ options to prevent participants from guessing answers. The responses were calculated as 1 for correct answers and 0 for incorrect or unknown answers. A total score of four subsections determined the overall nutrition knowledge score of the participant. 

#### 2.1.2. Content Validity

Content validity was calculated by item difficulty and discrimination. Each item should be neither too difficult nor too easy for participants. Kline [28] suggested that the difficulty of items should be between 20–80%. This criterion has been used in many previous studies to test the content validity of items [15,29]. Item discrimination is whether an item can effectively distinguish participants’ ability. While people with a high level of knowledge should answer it correctly, people with a low knowledge should not. Kline [28] suggested that the point-biserial correlation coefficient should be greater than 0.2. 

#### 2.1.3. Polit Study and Face Validity

Eleven participants were recruited, including a public nutritionist, a preventive medicine, and nine psychology graduate students. These participants were required to complete the C-GNKQ and evaluate its readability and the extent to which it measured nutrition knowledge. The data showed that most of the questions were highly readable, and a little lexical ambiguity was corrected according to participants’ suggestions. Participants expressed that these questions could measure their nutrition knowledge; the average score was 86.73 (1 to 100 points).

#### 2.1.4. Construct Validity

Construct validity indicates whether a nutrition knowledge questionnaire can measure nutrition knowledge or not [30]. The general nutrition knowledge score and the score for each factor should be higher for students majoring in nutrition than in other subjects [12]. In this study, nutrition or nutrition-related majors consist of food hygiene and nutrition, food science and engineering, food quality and safety, and preventive medicine. For these majors, nutrition education is often taken as a required course.

#### 2.1.5. Reliability

In order to test the reliability of the questionnaire, internal consistency and test-retest reliability were used as indicators. Internal consistency was determined by calculating the correlation coefficient of each item and factor score. The Cronbach’s ɑ should be greater than 0.7. The test-retest reliability was determined by calculating the correlation coefficient of the two tests from the same participant at two-week intervals. The correlation coefficient should also be greater than 0.7 [1].

### 2.2. Participants and Sample Size

The current study was conducted online with two participant samplings (*N* = 278). Sample A included 175 adults (47 males and 128 females) which were used to test internal consistency, content validity, and convergent validity. Those participants in sample A were recruited using quota sampling to make the sample a representation of the general population. However, sample B included 103 undergraduate students (50 students in nutrition-related majors and 53 students in nutrition-unrelated majors) which were recruited using convenience sampling. Therefore, the sample B only included young college students. It was mainly used to determine the construct validity and test different levels of nutrition knowledge among young people with different educations. Participants in sample B were also asked to complete the C-GNKQ twice at a two-week interval, and 71 of them were valid for assessing test-retest reliability. This study was conducted according to the guidelines laid out in the Declaration of Helsinki, and all procedures involving research study participants were approved by the Shaanxi Normal University ethics committee. All subjects were informed that the survey was not physically or mentally harmful to them, and they had the right to refuse or terminate the survey at any time. Verbal informed consent was obtained from all participants.

G-power was used to calculate the number of participants sufficient to measure construct validity [31], considering a two-tail independent sample t-test, using estimations of means and standard deviations from previous students’ data. A common value for the significant level (*ɑ* = 0.05) and a high statistical power (1–*β* = 0.95) were used. This study had a sample size larger than the recommended minimum of 42 participants per group to measure construct validity and get an effect size of *d* = 0.80.

### 2.3. Statistic Analysis

In this study, the data were analyzed with SPSS 25.0. The item difficulty was calculated by the number of participants who passed each item divided by all participants. The point-biserial correlation method was used to calculate the discrimination of each item, since they are dichotomous variables which required that the average total score of the participants that passed was significantly higher than that of the participants that did not pass. Independent sample t-test and one-way ANOVA were used to analyze the continuous variables (factor scores and total scores). Before the t-test and one-way ANOVA, Levene’s test for equality of variances was used to test the homogeneity of variance of the two groups. The Cronbach ɑ coefficient was used to estimate the internal consistency, and the Spearman correlation coefficient to assess the test-retest reliability. In the case of a significant one-way ANOVA, the LSD method was used to compare the difference between two levels of each variable. 

## 3. Results

### 3.1. Demographic Information

The characteristics of participants in sample A are shown in Table 1. A total of 175 questionnaires (82.5%) were valid, including 47 males and 128 females aged 18 to over 76 years old. Most of them had bachelor’s degrees or above (74.86%), and 58.29% had weight control behavior within three months (see Table 1). In addition, there were 103 valid participants in sample B (mean age = 21.91, *SD* = 1.70), including 31 males and 72 females. In total, 71 participants (16 males and 55 females, mean age = 22.28, *SD* = 1.75) in sample B completed the questionnaire twice to assess the test-retest reliability of C-GNKQ. 

### 3.2. Item Difficulty and Discrimination

The test version consists of 50 questions with 158 items. We first removed 18 questions with 90 items (4 questions with 17 items in *dietary recommendations*, 9 questions with 48 items in *source of nutrients*, 5 questions with 5 items in *daily food choice*, and 20 items in *diet–disease relationship*) because they did not meet the criteria of item difficulty or discrimination, as we had introduced in the section of content validity. The remaining items’ average difficulty was 59.36 (*SD* = 14.00), and the average discrimination was 0.33 (*SD* = 0.09). Therefore, the final version of C-GNKQ consisted of 32 questions with 68 items (see Appendix A) in categories of *dietary recommendations* (4 questions, 7 items); *source of nutrients* (16 questions, 38 items); *daily food choice* (3 questions, 3 items)*;* and *diet–disease relationship* (9 questions, 20 items). 

### 3.3. Construct Validity

In order to test whether C-GNKQ can accurately distinguish participants’ level of nutritional knowledge, students majoring in nutrition and other subjects were compared. In this study, 103 senior and graduate students were recruited, including 50 students majoring in nutrition (18 males and 32 females) and 53 students in other subjects (13 males and 40 females). The results showed that except for the *daily food choice* subscale, the other subscale scores and total scores of students majoring in nutrition were significantly higher than those majoring in other subjects, *p* < 0.001 (see Table 2).

### 3.4. Internal Consistency and Test-Retest Reliability

#### 3.4.1. Internal Consistency

The 175 samples were used to calculate Cronbach’s ɑ to measure the internal consistency reliability, and the data showed that the overall reliability of the scale was 0.885. The reliability coefficients of the four sub-dimensions were 0.563, 0.814, 0.334, and 0.771, respectively. However, the reliability of *dietary recommendations* and *daily food choice* subscales did not reach the reference of 0.7 (see Table 3). 

#### 3.4.2. Test-Retest Reliability

The overall test-retest reliability was 0.769, which reached the criterion of 0.7. The test-retest reliability of *dietary recommendations* and *source of nutrients* was 0.747 and 0.712, respectively. Nevertheless, the test-retest reliability of *daily food choice* and *diet-disease relationship* subscales was 0.495 and 0.593, respectively, so they did not reach 0.7 (see Table 3). Therefore, the overall scale has good reliability, but the subsection should be cautiously used alone. 

### 3.5. Convergent Validity

#### 3.5.1. Gender

There was a significant difference in the total score of C-GNKQ between males and females. The total score of females (*M* = 41.45, *SD* = 10.63) was significantly higher than that of males (*M* = 37.4, *SD* = 11.2), *t* = 2.201, *p* = 0.029. In the case of the four subscales, the score of females’ *sources of nutrients* (*M* = 22.85, *SD* = 6.36) was significantly higher than that of males (*M* = 20.38, *SD* = 6.32), *t* = 2.278, *p* = 0.024. There was no significant difference between males and females in other subscales (Figure 1). 

#### 3.5.2. Age

Due to the small number of participants over 56 years old, those over 46 years old were combined into the same group. There were significant differences in total score (*F*(3,171) = 2.723, *p* = 0.046), *dietary recommendations* (*F*(3,171) = 10.394, *p* < 0.001), and *sources of nutrients* (*F*(3,171) = 2.650, *p* = 0.050) among participants of different ages. However, the difference in *daily food choice* (*F*(3,171) < 1, *p* = 0.625) and *diet–diseases relationship* (*F*(3,171) < 1, *p* = 0.710) was not significant among participants in different age groups.

The post-test results showed that participants aged 18–25 (*M* = 42.49, *SD* = 10.80) scored significantly higher than those aged 26–35 (*M* = 37.41, *SD* = 10.67) and over 46 (*M* = 37.09, *SD* = 10.80) in the total score of the scale, *p* < 0.05. In terms of *dietary recommendations*, participants aged 18–25 (*M* = 4.31, *SD* = 1.75) were significantly higher than those aged 26–35 (*M* = 3.17, *SD* = 1.58), 36–45 (*M* = 2.73, *SD* = 1.28), and over 46 (*M* = 2.95, *SD* = 1.86), *p* < 0.001. In terms of *source of nutrients scores*, participants aged 18–25 (*M* = 23.40, *SD* = 6.47) scored significantly higher than those aged 26–35 (*M* = 20.17, *SD* = 6.74), *p* < 0.05 (Figure 2). 

#### 3.5.3. Marital Status

Due to the small number of divorced and widowed participants, the two groups were combined for data analysis. Participants with different marital statuses had significant differences in total score (*F*(2,172) = 4.527, *p* = 0.012), *dietary recommendations* (*F*(2,172) = 13.058, *p* < 0.001), and *sources of nutrients* (*F*(2,172) = 4.287, *p* = 0.015). There was no significant difference between the other subsections. 

The post-test results showed that the total scores of the divorced or widowed (*M* = 31.60, *SD* = 13.34) were significantly lower than those of the single (*M* = 41.80, *SD* = 11.10) and the married (*M* = 39.43, *SD* = 9.53), *p* < 0.05. In the matter of *dietary recommendations*, the scores of the single (*M* = 4.21, *SD* = 1.81) were significantly higher than those of the married (*M* = 2.95, *SD* = 1.41) and the divorced or widowed (*M* = 2.60, *SD* = 1.71), *p* < 0.001. Regarding *source of nutrients* score, the divorced or widowed (*M* = 17.20, SD = 7.24) scored significantly lower than the single (*M* = 23.03, *SD* = 6.45) and the married (*M* = 21.62, *SD* = 5.92), *p* < 0.05 (Figure 3).

#### 3.5.4. Income

The monthly income data were combined into three groups as follows: less than 2000, 2000 to 5000, and more than 5000. Participants with different incomes had significant differences in *dietary recommendations* (*F*(3,162) = 2.843, *p* = 0.040) and *diet–diseases relationship* (*F*(3,162) = 2.743, *p* = 0.045). There was no significant difference in the total score (*F*(3,162) = 1.762, *p* = 0.157), *source of nutrients* (*F*(3,162) = 2.111, *p* = 0.101), and *daily food choice* (*F*(3,162) < 1, *p* = 0.969).

The post-test results showed that *dietary recommendation* scores of participants with a monthly income less than 2000 RMB (*M* = 4.11, *SD* = 1.75) were significantly higher than those with a monthly income between 2000 and 5000 RMB (*M* = 3.32, *SD* = 1.69), *p* < 0.01. However, in the *diet–disease relationship*, scores of participants with a monthly income of 2000 to 5000 RMB (*M* = 13.49, *SD* = 3.74) were significantly higher than those with a monthly income less than 2000 RMB (*M* = 11.97, *SD* = 3.67) and more than 5000 RMB (*M* = 11.22, *SD* = 4.21), *p* < 0.05 (Figure 4).

#### 3.5.5. Education Level

Since there were fewer participants with primary and junior high school education, they were combined together for data analysis. The results showed that there were significant differences in the total score (*F*(3,171) = 7.409, *p* < 0.001), *dietary recommendations* (*F*(3,171) = 6.859, *p* < 0.001), *sources of nutrients* (*F*(3,171) = 5.746, *p* = 0.001), *daily food choice* (*F*(3,171) = 2.247, *p* = 0.018), and *diet–diseases relationship* (*F*(3,171) = 4.473, *p* = 0.005). 

The post-test results showed that the total score of the group with a junior high school degree or below (*M* = 25.40, *SD* = 9.07) was significantly lower than the group with senior high school (*M* = 41.32, *SD* = 11.01), undergraduate (*M* = 41.36, *SD* = 9.99), and master’s degree or above (*M* = 40.94, *SD* = 11.06), *p* <0.001. 

In terms of *dietary recommendations*, the scores of the group with a junior high school degree or below (*M* = 1.30, *SD* = 0.95) were significantly lower than the group with senior high school (*M* = 3.79, *SD* = 1.86), undergraduate (*M* = 3.78, *SD* = 1.79), and master’s degree or above (*M* = 3.91, *SD* = 1.40), *p* <0.001. 

On the *source of nutrients*, the score of the group with a junior high school degree or below (*M* = 14.40, *SD* = 4.99) was significantly lower than the group with high school (*M* = 22.24, *SD* = 6.66), undergraduate (*M* = 22.64, *SD* = 5.94), and master’s degree or above (*M* = 23.19, *SD* = 6.68), *p* < 0.001. 

In terms of *daily food choice*, the scores of the group with junior high school degree or below (*M* = 1.60, *SD* = 0.97) were significantly lower than the group with senior high school (*M* = 2.53, *SD* = 0.75), undergraduate (*M* = 2.33, *SD* = 0.76), and master’s degree or above (*M* = 2.28, *SD* = 0.96), *p* < 0.05. 

In terms of *diet–disease relationship*, the scores of the group with junior high school degree or below (*M* = 8.10, *SD* = 4.18) were significantly lower than the group with high school (*M* = 12.76, *SD* = 3.87), undergraduate (*M* = 12.62, *SD* = 3.93), and master’s degree or above (*M* = 11.56, *SD* = 4.06), *p* < 0.05 (Figure 5).

#### 3.5.6. Weight Control in Three Months

The results of an independent sample t-test showed that there was no significant difference between total score (*t* (173) < 1, *p* = 0.826), *dietary recommendations* (*t* (173) < 1, *p* = 0.528), *sources of nutrients* (*t* (173) < 1, *p* = 0.985), *daily food choice* (*t* (173) < 1, *p* = 0.801), and *diet–diseases relationship* (*t* (173) < 1, *p* = 0.816) between different weight control groups. 

## 4. Discussion

This study aimed to develop a Chinese version of GNKQ. The data showed that the revised Chinese version of C-GNKQ, which provided an effective tool to measure nutrition knowledge for Chinese adults, had good reliability and validity with a shorter length.

### 4.1. Internal Consistency and Construct Validity 

The internal consistency coefficient of the C-GNKQ questionnaire was 0.885, between 0.70 [12] and 0.92 [13]. However, the internal consistency of the *dietary recommendations* subscale and *daily food choice* subscale was 0.563 and 0.334, less than the 0.7 criteria, and these data were consistent with previous studies [14,25]. For example, the Cronbach’s ɑ values of *dietary recommendations* and *daily food choice* in the Italian version of GNKQ were 0.12 and 0.40, respectively. The lower internal consistency for the subscales of dietary recommendations and daily food choice may be influenced by the currently used Chinese dietary guidelines. The Chinese dietary guidelines show that the Chinese dietary patterns are different from western countries [18,19,31]. For example, the intake of processed meat and sugar-sweetened beverages is lower in China, so it is not a significant problem in China. In addition, the number of items in the two subscales was reduced in the final version of C-GNKQ, which could affect the size of the Cronbach’s ɑ values [32].

The construct validity was measured between the participants majoring in nutrition or not. The data showed that the total scores and scores of other subscales except for *daily food choice* subscale scores were significantly larger for the participants majoring in nutrition than those majoring in nutrition-unrelated fields. It seemed that the smaller number of subscale items led to a lower Cronbach’s ɑ value, since the subscales of *dietary recommendations* and *daily food choice* only included seven and three items, respectively. The data meant that the two subscales could not be used alone, but it did not influence the use of the overall scale. 

Compared with the Turkish version of GNKQ, the C-GNKQ had the following changes. First, we removed some items based on dietary habits in China. For example, as an unfamiliar food type, butter is rarely consumed by Chinese, so we removed the items on fat and calorie content in butter in *source of nutrients*. Similarly, we removed the items on pudding and cheese in *daily food choice* section. Second, we revised some items according to the Chinese Dietary Guidelines. For example, in the *dietary recommendations* section, we added the item about fat as one of the essential nutrients, the question on types of energy-supplying nutrients, and the recommended ratio of calorie intake for each meal. Finally, we removed some items with low item difficulty or discrimination. These items included salt content in foods, the relationship between fat and cholesterol, and the difference between brown sugar and white sugar. Although the modifications may affect the direct comparison of nutrition knowledge across cultures, the adaption was necessary. Overall, the new version can better reflect the nutrition knowledge of Chinese people, as evidenced by the reliability and validity of the C-GNKQ.

### 4.2. Test-Retest Reliability

The test-retest reliability was calculated between two tests from the same participants at two-week intervals. The data showed that the coefficients of the total score of C-GNKQ and the subscales of *dietary recommendations* and *sources of nutrients* were higher than 0.7, consistent with the criterion of good reliability. However, the reliability of the *daily food choice* did not reach 0.7, which might be influenced by the fewer items, consistent with previous studies [15,17]. The Spearman correlation coefficient of the *diet–disease relationship* subscale was 0.593, which reached moderate reliability [1]. 

### 4.3. Convergent Validity

The nutritional knowledge level was higher in women than men and lower in participants with junior high school degrees compared with those with high school degrees or above. These results were consistent with previous studies [6,33,34]. It has been previously shown that nutritional knowledge level was negatively correlated with age and positively correlated with income level [35]. However, this study found that the nutritional knowledge score of participants between 18 to 25 years old was significantly higher than the score of participants between 26 to 35 years old. Moreover, the nutritional knowledge score of participants with a monthly income of less than 2000 RMB was significantly higher than those with a monthly income between 2000 and 5000 RMB. 

These results were inconsistent with previous studies in western countries [7]. It might be because young participants have a higher education in the Chinese context. In this study, participants between 18 to 25 years old were mostly college students or graduate students, and their monthly income was less than 2000 RMB. Still, they have higher education levels (undergraduate or master) than older participants, resulting in different outcomes than previous studies. 

## 5. Limitation and Future Directions

This study has certain limitations. For instance, the sample was relatively small and does not include enough representative populations. As in previous studies, many demographic variables showed a significant difference in nutrition knowledge level. Although sample A included some older participants, most of them were young adults with a higher level of education. Future studies could recruit participants from different populations to test and expand the use of the C-GNKQ in China. Moreover, it should be noticed that due to the limitation of participant sampling, the results have relatively better reliability and validity for younger individuals with a higher level of education. 

In addition, the data have significant value for nutrition education in the Chinese context; however, on some subscales, C-GNKQ had lower reliability and validity, which should be cautiously used alone in future studies. The reliability and validity of subscales, such as *dietary recommendations* and *daily food choices*, should be investigated in further studies. In the present study, in order to balance the sociodemographic differences between the two groups used the construct validity test, we compared college students majoring in nutrition-related and students majoring in nutrition-unrelated subjects to test the construct validity of the questionnaire, but perhaps replacing the students majoring in nutrition with qualified health experts would make the results more convincing. In addition, because sample B only included college students, the results of structural validity and retest reliability may be more applicable to the young and educated population. It should be remembered that food choices are influenced by the interactions among humans-, food-, and environment-related variables. Therefore, it is difficult to predict while both conscious and unconscious thinking are involved in the process [36]. Despite the above limitations, the overall C-GNKQ has good reliability and validity, which can fill the gap of a systematically designed nutrition knowledge questionnaire used in China. 

## 6. Conclusions

In conclusion, the first Chinese version of the general nutrition knowledge questionnaire (C-GNKQ) was developed in this study. This C-GNKQ has enough power to effectively test the nutrition knowledge level of Chinese adults, which can potentially play a role in preventing food-related chronic diseases, such as obesity, in the future. 

## Figures and Tables

**Figure 1 nutrients-13-04353-f001:**
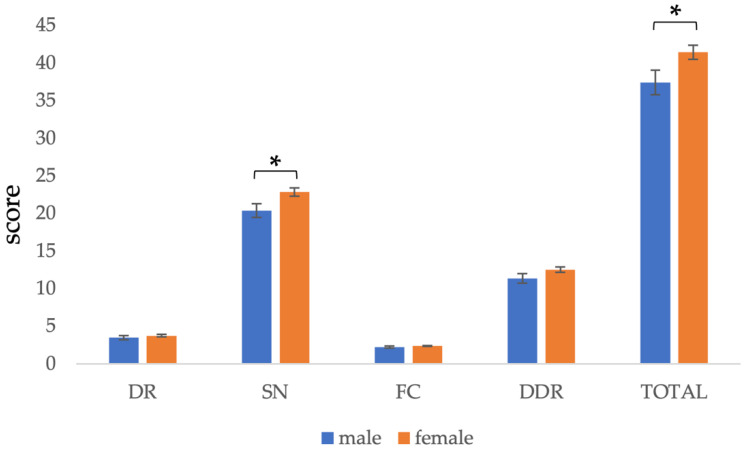
Comparison of C-GNKQ scores between males and females. Error bars represent standard errors. * indicates *p* < 0.05.DR: dietary recommendations, SN: sources of nutrients, FC: daily food choice, and DDR: diet–diseases relationship.

**Figure 2 nutrients-13-04353-f002:**
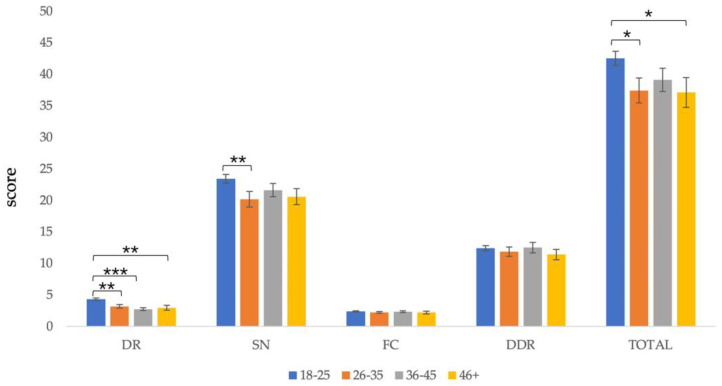
Comparison of C-GNKQ scores between different age groups. Error bars represent standard errors. * indicates *p* < 0.05, ** indicates *p* < 0.01, and *** indicates *p* < 0.001. DR: dietary recommendations, SN: sources of nutrients, FC: daily food choice, and DDR: diet–diseases relationship.

**Figure 3 nutrients-13-04353-f003:**
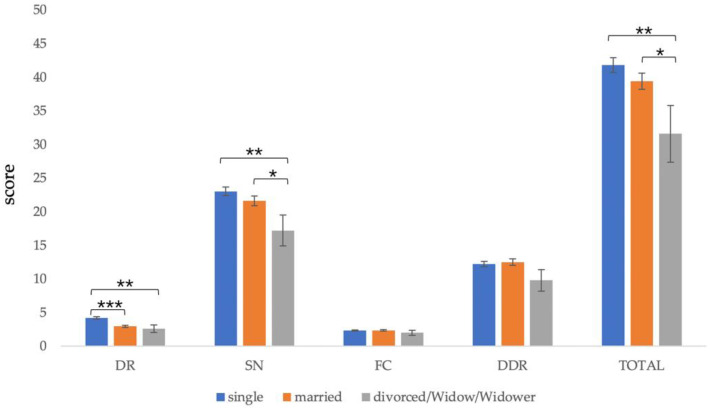
Comparison of C-GNKQ scores between different marital statuses. Error bars represent standard errors. * indicates *p* < 0.05, ** indicates *p* < 0.01, and *** indicates *p* < 0.001. DR: dietary recommendations, SN: sources of nutrients, FC: daily food choice, and DDR: diet–diseases relationship.

**Figure 4 nutrients-13-04353-f004:**
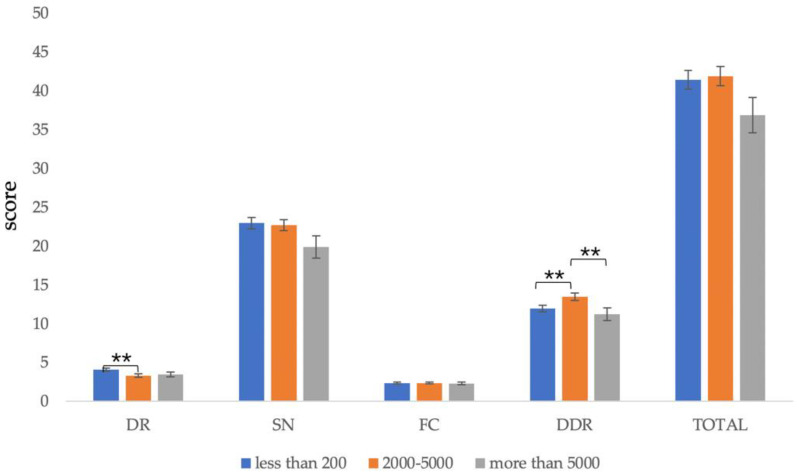
Comparison of C-GNKQ scores between different levels of income groups. Error bars represent standard errors. ** indicates *p* < 0.01. DR: dietary recommendations, SN: sources of nutrients, FC: daily food choice, and DDR: diet–diseases relationship.

**Figure 5 nutrients-13-04353-f005:**
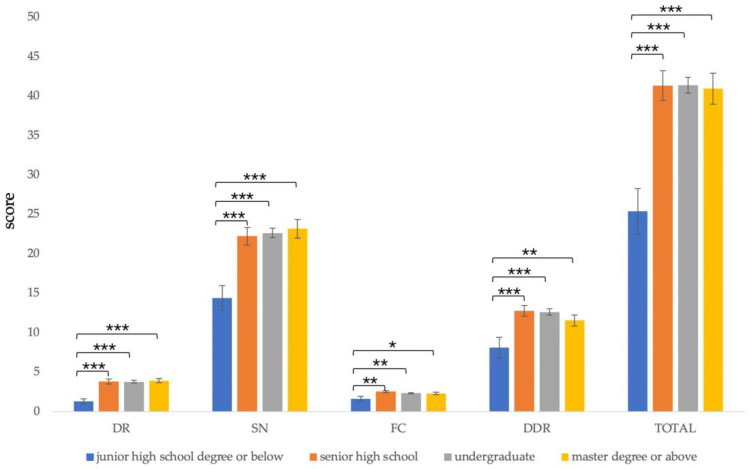
Comparison of C-GNKQ scores between different levels of education groups. Error bars represent standard errors. * indicates *p* < 0.05, ** indicates *p* < 0.01, and *** indicates *p* < 0.001. DR: dietary recommendations, SN: sources of nutrients, FC: daily food choice, and DDR: diet–diseases relationship.

**Table 1 nutrients-13-04353-t001:** Demographic characteristics of the participants in sample A (*N* = 175).

Demographic Variables	*N*	Demographic Variables	*N*
**Gender**		**Income (RMB/month)**	
Male	47	<2000	73
Female	128	2001–5000	59
**Age (y)**		5001–10,000	18
18–25	89	10,001–50,000	7
26–35	31	>50,001	2
36–45	33	No reported	16
46–55	18	**Education level**	
56–65	1	Primary school graduate	4
66–75	0	Junior school graduate	8
>76	3	High school graduate	34
**Marital status**		University degree	99
Single	101	Graduate degree	32
Married	64	**Weight control in 3 months**	
Divorced	8	Yes	73
Widow/Widower	2	No	102

*Note*: RMB: Chinese yuan.

**Table 2 nutrients-13-04353-t002:** Comparison of C-GNKQ scores between participants with different nutrition education.

	Students Majoring in Nutrition (*n* = 50)		Students Majoring in Other Subjects (*n* = 53)		*t* Test	
Subsection (number of items)	*M*	*SD*	*M*	*SD*	*t*	*p*
Dietary Recommendations (7)	5.90	1.33	3.62	2.06	6.707	<0.001
Source of Nutrients (38)	31.08	4.52	25.49	6.81	4.879	<0.001
Daily Food Choice (3)	2.38	0.70	2.55	0.70	−1.218	0.226
Diet–disease Relationship (20)	15.68	3.01	12.72	3.58	4.534	<0.001
Total (68)	55.04	6.87	44.38	11.16	5.874	<0.001

**Table 3 nutrients-13-04353-t003:** Internal consistency and test-retest reliability of the revised C-GNKQ.

Subsection (Number of Items)	Cronbach’s *ɑ*	Spearman Correlation Coefficient	*p*
Dietary Recommendations (7)	0.563	0.747	<0.001
Source of Nutrients (38)	0.814	0.712	<0.001
Daily Food Choice (3)	0.334	0.495	<0.001
Diet–disease Relationship (20)	0.771	0.593	<0.001
Total (68)	0.885	0.769	<0.001

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
