# Peer review of "Development and Validity of a General Nutrition Knowledge Questionnaire (GNKQ) for Chinese Adults"

_nutrients, 2021, doi:10.3390/nu13124353_

Round 1

Reviewer 1 Report

Dear Authors, well done on completing this research project and resulting publication. I agree that your paper has addressed an important issue. 

My feedback is split into two parts, general comments and specific comments. 

A) General feedback

  • The paper addresses that the Chinese population does not yet have a standardised and validated measure of level of nutrition knowledge. This is an identified gap. 
  • Use of the traditional GNKQ is admirable. I like the research into the Chinese dietary guidelines and how they should be incorporated into the revised tool. 
  • Validation was necessary and undertaken. What was lacking for me was more evidence about the link between the modifications and the dietary guidelines themselves, justification for why those modifications were done and what impact they could have on measuring knowledge. What is the intended use of the survey and how could it be of benefit to those who would use it? 
  • Some of the grammar in the paper needs revising to reach an international audience. I acknowledge that this is a minor correction in the scope of the paper. 
  • Addition of questions to the traditional scaffold of the GNKQ will impact upon its reliability, how widely the results can be compared to other cohorts across the world. I would have liked this to be discussed in more depth in the paper. 
  • Use of dietetics students and non-dietetics students is a good step in the right direction for the validity testing, but I would have liked to see this also done by qualified health experts who are not in training and may not have inaccurate levels of nutrition knowledge whilst still undergoing training. I acknowledge that the testing has already taken place, perhaps this could be a recognised limitation instead at this point. 
  • Choice of desired accuracy according to Kline's 20-80% level of difficulty needed more explanation and justification. Some of the decisions in this paper needed more clarity. For academics like me who would be very interested in seeing results of it in use and framing recommendations on it or comparing results, we need as much evidence and justification behind the choices in constructing the tool to help interpret the impact of the results in the future. 
  • When interpreting the results and differences in findings, I would have liked to see more reflection on the potential reasons for these differences. It will help future users of this test/tool account for them or understand whether or not the tool is appropriate for their use. 
  • Limitations were quite short and they were a little glossed over. 
  • I was wondering if the funding agency will have any input in the rollout or wider use of the survey once it has been developed and piloted? This was not clear in the financial support section. 
  • Have you looked at other authors who have used the GNKQ in the Western settings? There were a couple of papers that I did not see utilised that you may benefit from digging out to read the limitations of use and how the questions may have been changed or used. This would boost your literature review section and justification of use. 

B) Specific comments

  • Abstract, line 10-11. "However, due to the lack 10
    of nutrition knowledge and unhealthy eating patterns, more and more Chinese people face obesity." I would caution in simplifying obesity to these issues only. Suggestion: rewrite sentence to be more encompassing. 
  • Abstract: I hope to read about the validity testing and choices of groups and types of validity being tested, eg why were the experts used compared to students. 
  • Lines 30-32: Nutritional knowledge is important, but why do the list of your eating patterns determine healthy eating? I would recommend here that you mean for the general population and according to whom these recommendations are 'healthy'
  • Line 34: Stroke is not a chronic disease state, cardiovascular disease is. 
  • Line 42-43 I would rewrite " there has been no systematic study to" as However, as yet a nutrition knowledge questionnaire has not been systematically studied or created for Chinese people. 
  • Line 60-62.  How are Chinese eating patterns different from Western ones? It would help to spell these out as your audience may not be familiar with the differences. 
  • Lines 71-73 is repetitive from the section above. 
  • Line 78: Please explain the consistencies between the Turkish eating guidelines and the Chinese guidelines.
  • Line 87: Brief explanation of the Declaration of Helsinki guidelines utilised or applicable to benefit the audience please. 
  • Section 2.1.1: This section is quite important and sets aside the traditional GNKQ to your modified version. Bear in mind what this means for replication, interpretation and comparison between your tool and the original tool. I hope that this is raised in the limitations or discussion section. 
  • Line 109: Suggested term for "high knowledge" is a "high level of knowledge" which is more grammatically correct. Same for low knowledge, as low level of knowledge. 
  • Line 121: Reference for construct validity explanation?
  • Line 125: Please do not write "etc" in a peer reviewed article. I would recommend listing the majors or condensing the list. I am also unclear on how many of these majors actually involve nutrition education if they "often" take it. This is too broad a term.
  • Line 129-130: Reference for reliability please. 
  • Line 160: What does "weight control behaviour" mean and why is this important to measure in the demographics? How were participants asked this exactly in the survey?
  • Table 1: Is "Income (RMB/mouth)" a typo instead of RMB/month?
  • Table 1: Was there a reason that gender was measured in binary only?
  • Line 158: 175 surveys were valid. What does this mean? Completed? 
  • Line 162 - 166. I am unclear in this section how many questions were removed and this is important. I believe that the following sentence details how many were remaining and therefore that this beginning sentence details how many were removed. I had to re-read this a few times to get that information and this was therefore not clear. 
  • Section 3.5. I understand that the results in this section help explain the convergent validity. I query whether or not the information could be better represented for each of the categories to summarise it, rather than being text based?
  • I see that the comparison of results with other countries' studies like the Italian study mean that the use of the survey is described as still favourable. I feel that this is likening apples and oranges together, so this could be detailed in the limitations section instead of seen as a strength. 
  • Line 311: Significant difference between 18-25 yo and 26-36yo - by how much was this different? What significance level? Why do you think this was? 

Author Response

Dear the Reviewers,

Thank you very much for the valuable comments. We have carefully revised the manuscript to improve it based on your comments.Some of the major changes made in the revised manuscript have been highlighted in green. The highlighted sections do not include the deleted contents and revisions made on language problems

Reviewer 2 Report

This was an ambitious and important study addressing the development and validity of a general nutrition knowledge questionnaire. The manuscript is overall well written, but the methods and results section need to be improved. Unfortunately, the study has several methodological limitations. The stratified analysis presented in the results section under convergent validity is not optimal due to the skewed selection of participants. See comments below. 

The first section of the 2.1. paragraph is repetitive to second to last section in the introduction. Consider to rewrite. The paragraph 2.1 must be rewritten as it is uncleared when the authors are referring to the present or previous studies. Further clarification on how the Turkish version was selected and changed is needed. 

Why did the researchers add 11 questions to the Turkish GNKQ? Why were specifically these 11 questions added? What are the questions? This needs to be incorporated in the manuscript. The full version of the C-GNKQ is missing and it is not clear which parts that was amended. Provide the full questionnaire as supplementary material. It is also unclear in which sections these questions were added, and how tis affected the validity. Were the 11 added questions in the sections with lower internal validity? How was the validity of these sections without the new questions? 

Under section 2.1.2 the researchers write that content validity was calculated, what was the content validity? I cannot find the results from this validity check. 

The selection of participants in this study is a major limitation. I do not find the stratified analysis of demographic variables meaningful because of the skewed sample. First of all, the number of participants in the study does not add up. Therefore, it is not possible for the reviewer to interpret the results correctly. In Table 1, the authors report 175 participants, however in the results section, there is only 103 individuals reported. Table 1 should be stratified for individuals majoring in nutrition and not, as these are the two comparison groups. It is also unclear which participants did the test two times. Where are these participants and results presented? How were these participants selected? How long time past between the tests? This needs to be much clearer. Second, a vast majority of individuals are women. Are the comparison of men and women of meaning? Can the results be generalized to men?

Why were mainly university degree persons involved in the study? As the most limited knowledge in nutrition are among those with lower levels of education it would be important to test the questionnaire also aiming those individuals. This should be stated in the method section, a description of why this sample was made, and how it came to be mostly individuals with higher level of education. In addition, this should be thoroughly stated under strengths and limitations section. Because this limit the results of the study considerable, it should also be included in the purpose, as the results from this study cannot be generalized to the general population, e.g.; “ … and test its reliability and validity on younger individuals with higher level of education”.

I recommend that the author creates a new table, incorporating the results from the convergent validity in stratified demographic variables. Without a table, the results are really hard to follow overview.  

The results show that younger, female with lower income level had higher score on the C-GNKQ. However, as previously stated, the characteristics of the students with a major in nutrition is not described. It is impossible for the reviewer and reader to determent which factors that are affecting the results as the description of how the participants were selected, and the description of the participants in the study is lacking. I believe that the manuscript needs major revision before undergoing critical review. The authors need to restructure the methods and results section of the manuscript. The discussion is artificial. The authors do not provide any reasoning to why their results differ to this extent compared with previous validation studies of the questionnaire. Without a proper structured selection of participants and a proper well-structured results section, one cannot determine whether the results are valid. This is unfortunate as the aim of the study is important and one can tell that the authors have made an ambitious attempt of testing the questionnaire.

Author Response

Dear the Reviewers,

Thank you very much for the valuable comments. We have carefully revised the manuscript to improve it based on your comments.Some of the major changes made in the revised manuscript have been highlighted in green. The highlighted sections do not include the deleted contents and revisions made on language problems.

Round 2

Reviewer 2 Report

Comment 4

The selection of participants in this study is a major limitation. I do not find the stratified analysis of demographic variables meaningful because of the skewed sample. First of all, the reviewer to interpret the results correctly. In Table 1, the authors report 175 participants, however in the results section, there is only 103 individuals reported. Table 1 should be stratified for individuals majoring in nutrition and not, as these are the two comparison groups. It is also unclear which participants did the test two times. Where are these participants and results presented? How were these participants selected? How long time past between the tests? This needs to be much clearer. Second, a vast majority of individuals are women. Are the comparison of men and women of meaning? Can the results be generalized to men?

Reply

We agree with that about the lack of representativeness of the sample. This study is the first attempt to revise the GNKQ for the Chinese population. Since the questionnaire is too long, it was difficult for us to collect subjects that were sufficiently representative of the entire Chinese adult population for this study. We also illustrated this limitation, please see lines 400-405.

As in previous studies, many demographic variables showed a significant difference in nutrition knowledge level. Future studies could recruit participants from different populations to make tests and expand the use of the C-GNKQ in China. Moreover, it should be noticed that, due to the limitation of participants sampling, the results have better reliability and validity on younger individuals with a higher level of education.

In the present study, we recruited two independent samples comprising 175 and 103 participants, respectively. The first sample included 175 participants to determine internal consistency, content validity, and convergent validity. All participants completed a questionnaire only once. The second sample included 103 students (50 students majoring in nutrition, 53 students majoring in other subjects) to determine to construct validity and test-retest reliability. Participants in the second sample were asked to complete a questionnaire twice at two-week intervals.

Reviewer reply:

These clarifications have made it easier to follow the logic of the manuscript. However, this is still not clear when reading the manuscript. I believe readers will be confused by the different sampling and two steps of the study without any clarification of this in the present manuscript. The description of the two different samples, and how the participants in these two samples were selected is still missing.

The aim of the study was to “develop a General Nutrition Knowledge Questionnaire for Chinese adults 79 (C-GNKQ) and test its reliability and validity”. This aim does not include the second part of your study, as the authors state: “to test whether C-GNKO can accurately distinguish participants´ level of nutritional knowledge”. To me, this is another type of aim and study.

In addition, as I know understand it, the analysis of the demographic variables was only for the first 175 individuals. What was the demographics of the second sample? Why was this not included in the study? The authors state that it was important to determine demographic differences of participants, then this should be made for the whole sample.

Although the sample in this study was not evenly split between males and females, since the focus of this study was to revise the nutrition knowledge questionnaire, we thought it was necessary to report gender differences to let the readers or other researchers know the potential factors might influence the differences in results. Secondly, we also acknowledge that the results may be difficult to generalize to all Chinese populations and therefore add this note to the limitations of the manuscript. Please see lines 403-405.

Moreover, it should be noticed that, due to the limitation of participants sampling, the results have better reliability and validity on younger individuals with a higher level of education.

Comment 5

Why were mainly university degrees involved in the studies? As the most limited knowledge of nutrition is among those with lower levels of education, it would be important to test the questionnaire also aiming at those individuals. This should be stated in the method section, a description of why this sample was made, and how it came to be mostly individuals with a higher level of education. In addition, this should be thoroughly stated under the strengths and limitations section. Because this limits the results of the study , it should also be included in the purpose, as the results from this study cannot be generalized to the general population, e.g.: “… and test its reliability and validity on younger individuals with higher level of education”.

Reply

Thanks for your suggestion. As mentioned above, we recruited two independent samples to determine the reliability and validity of C-GNKQ. In the present study, to try to balance the sociodemographic differences between groups in the construct validity test, we compared students majoring in nutrition and students majoring in other subjects to test the construct validity of the questionnaire. However, it indeed include 56% of participants with a bachelor's degree in the first sample. We have added it as a limitation in the manuscript,please see lines 403-405.

Moreover, it should be noticed that, due to the limitation of participants sampling, the results have relatively better reliability and validity on younger individuals with higher level of education.

Reviewer reply:

I believe that this amendment is not sufficient. As the reviewers had the intention of recruiting participants at a university, I still believe that this should be brought up in the method section, and included in the aim of the study.

Comment 6

I recommend that the author creates a new table, incorporating the results from the convergent validity in stratified demographic variables. Without a table, the results are really hard to follow overview. 

Reply

Thanks for your suggestions. We have added figures 1-5 to describe the results of convergent validity.

Reviewer reply:

The figures make the results much clearer. Please add abbreviations in the figure legends.

Comment 7

The results show that younger, female with lower income level had higher score on the C-GNKQ. However, as previously stated, the characteristics of the students with a major in nutrition is not described. It is impossible for the reviewer and reader to determent which factors that are affecting the results as the description of how the participants were selected, and the description of the participants in the study is lacking. I believe that the manuscript needs major revision before undergoing critical review. The authors need to restructure the methods and results section of the manuscript. The discussion is artificial. The authors do not provide any reasoning to why their results differ to this extent compared with previous validation studies of the questionnaire. Without a proper structured selection of participants and a proper well-structured results section, one cannot determine whether the results are valid. This is unfortunate as the aim of the study is important and one can tell that the authors have made an ambitious attempt of testing the questionnaire.

Reply

Thanks for your suggestions. As mentioned above, the data of the first sample including 175 adults was used to determine internal consistency, content validity, and convergent validity. And the characteristics of the first sample were shown in Table 1.

Reviewer reply:

As stated above, the amendments made by the authors does still not clarify my above addressed concerns. I believe readers will be confused by the different sampling and two steps of the study without any clarification of this in the present manuscript. The description of the two different samples, and how the participants in these two samples were selected is still missing. Why is the characteristics of the second sample not included?

Author Response

Dear Reviewer,
Thank you very much for the valuable comments. We have carefully revised the manuscript to improve it based on your comments.Some of the major changes made in the revised manuscript have been highlighted in yellow. The highlighted sections do not include the deleted contents and revisions made on language problems.
